# Question Asking as Program Generation

**Anselm Rothe**[1]
anselm@nyu.edu

**Brenden M. Lake**[1,2]
brenden@nyu.edu

**Todd M. Gureckis**[1]
todd.gureckis@nyu.edu

[1]Department of Psychology    [2]Center for Data Science
New York University

## Abstract

A hallmark of human intelligence is the ability to ask rich, creative, and revealing questions. Here we introduce a cognitive model capable of constructing human-like questions. Our approach treats questions as formal programs that, when executed on the state of the world, output an answer. The model specifies a probability distribution over a complex, compositional space of programs, favoring concise programs that help the agent learn in the current context. We evaluate our approach by modeling the types of open-ended questions generated by humans who were attempting to learn about an ambiguous situation in a game. We find that our model predicts what questions people will ask, and can creatively produce novel questions that were not present in the training set. In addition, we compare a number of model variants, finding that both question informativeness and complexity are important for producing human-like questions.

## 1  Introduction

In active machine learning, a learner is able to query an oracle in order to obtain information that is expected to improve performance. Theoretical and empirical results show that active learning can speed acquisition for a variety of learning tasks [see 21, for a review]. Although impressive, most work on active machine learning has focused on relatively simple types of information requests (most often a request for a supervised label). In contrast, humans often learn by asking far richer questions which more directly target the critical parameters in a learning task. A human child might ask "Do all dogs have long tails?" or "What is the difference between cats and dogs?" [2]. A long term goal of artificial intelligence (AI) is to develop algorithms with a similar capacity to learn by asking rich questions. Our premise is that we can make progress toward this goal by better understanding human question asking abilities in computational terms [cf. 8].

To that end, in this paper, we propose a new computational framework that explains how people construct rich and interesting queries within in a particular domain. A key insight is to model questions as programs that, when executed on the state of a possible world, output an answer. For example, a program corresponding to "Does John prefer coffee to tea?" would return `True` for all possible world states where this is the correct answer and `False` for all others. Other questions may return different types of answers. For example "How many sugars does John take in his coffee?" would return a number 0, 1, 2, etc. depending on the world state. Thinking of questions as syntactically well-formed programs recasts the problem of question asking as one of *program synthesis*. We show that this powerful formalism offers a new approach to modeling question asking in humans and may eventually enable more human-like question asking in machines.

We evaluate our model using a data set containing natural language questions asked by human participants in an information-search game [19]. Given an ambiguous situation or context, our model can predict what questions human learners will ask by capturing constraints in how humans construct semantically meaningful questions. The method successfully predicts the frequencies of

human questions given a game context, and can also synthesize novel human-like questions that were not present in the training set.

## 2 Related work

Contemporary active learning algorithms can query for labels or causal interventions [21], but they lack the representational capacity to consider a richer range of queries, including those expressed in natural language. AI dialog systems are designed to ask questions, yet these systems are still far from achieving human-like question asking. Goal-directed dialog systems [25, 1], applied to tasks such as booking a table at a restaurant, typically choose between a relatively small set of canned questions (e.g., "How can I help you?", "What type of food are you looking for?"), with little genuine flexibility or creativity. Deep learning systems have also been developed for visual "20 questions" style tasks [22]; although these models can produce new questions, the questions typically take a stereotyped form ("Is it a person?", "Is it a glove?" etc.). More open-ended question asking can be achieved by non-goal-driven systems trained on large amounts of natural language dialog, such as the recent progress demonstrated in [20]. However, these approaches cannot capture intentional, goal-directed forms of human question asking.

Recent work has probed other aspects of question asking. The Visual Question Generation (VQG) data set [16] contains images paired with interesting, human-generated questions. For instance, an image of a car wreck might be paired with the question, "What caused the accident?" Deep neural networks, similar to those used for image captioning, are capable of producing these types of questions after extensive training [16, 23, 11]. However, they require large datasets of images paired with questions, whereas people can ask intelligent questions in a novel scenario with no (or very limited) practice, as shown in our task below. Moreover, human question asking is robust to changes in task and goals, while state-of-the-art neural networks do not generalize flexibly in these ways.

## 3 The question data set

Our goal was to develop a model of context-sensitive, goal-directed question asking in humans, which falls outside the capabilities of the systems described above. We focused our analysis on a data set we collected in [19], which consists of 605 natural language questions asked by 40 human players to resolve an ambiguous game situation (similar to "Battleship").[1] Players were individually presented with a game board consisting of a 6×6 grid of tiles. The tiles were initially turned over but each could be flipped to reveal an underlying color. The player's goal was to identify as quickly as possible the size, orientation, and position of "ships" (i.e., objects composed of multiple adjacent tiles of the same color) [7]. Every board had exactly three ships which were placed in non-overlapping but otherwise random locations. The ships were identified by their color $S = \{Blue, Red, Purple\}$. All ships had a width of 1, a length of $N = \{2, 3, 4\}$ and orientation $O = \{Horizontal, Vertical\}$. Any tile that did not overlap with a ship displayed a null "water" color (light gray) when flipped.

After extensive instructions about the rules and purpose of the game and a number of practice rounds [see 19], on each of 18 target contexts players were presented with a partly revealed game board (similar to Figure 1B and 1C) that provided ambiguous information about the actual shape and location of the ships. They were then given the chance to ask a natural-language question about the configuration. The player's goal was to use this question asking opportunity to gain as much information as possible about the hidden game board configuration. The only rules given to players about questions was that they must be answerable using one word (e.g., true/false, a number, a color, a coordinate like A1 or a row or column number) and no combination of questions was allowed. The questions were recorded via an HTML text box in which people typed what they wanted to ask. A good question for the context in Figure 1B is "Do the purple and the red ship touch?", while "What is the color of tile A1?" is not helpful because it can be inferred from the revealed game board and the rules of the game (ship sizes, etc.) that the answer is "Water" (see Figure 3 for additional example questions).

Each player completed 18 contexts where each presented a different underlying game board and partially revealed pattern. Since the usefulness of asking a question depends on the context, the data

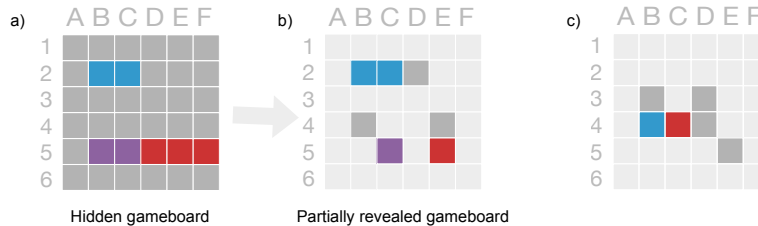

Figure 1: The Battleship game used to obtain the question data set by Rothe et al. [19]. (A) The hidden positions of three ships $S = \{Blue, Red, Purple\}$ on a game board that players sought to identify. (B) After observing the partly revealed board, players were allowed to ask a natural language question. (C) The partly revealed board in context 4.

set consists of 605 question-context pairs $\langle q, c \rangle$, with 26 to 39 questions per context.[2] The basic challenge for our active learning method is to predict which question $q$ a human will ask from the given context $c$ and the overall rules of the game. This is a particularly challenging data set to model because of the the subtle differences between contexts that determine if a question is potentially useful along with the open-ended nature of human question asking.

## 4 A probabilistic model of question generation

Here we describe the components of our probabilistic model of question generation. Section 4.1 describes two key elements of our approach, compositionality and computability, as reflected in the choice to model questions as programs. Section 4.2 describes a grammar that defines the space of allowable questions/programs. Section 4.3 specifies a probabilistic generative model for sampling context-sensitive, relevant programs from this space. The remaining sections cover optimization, the program features, and alternative models (Sections 4.4-4.6).

### 4.1 Compositionality and computability

The analysis of the data set [19] revealed that many of the questions in the data set share similar concepts organized in different ways. For example, the concept of ship size appeared in various ways across questions:

- "How long is the blue ship?"
- "Does the blue ship have 3 tiles?"
- "Are there any ships with 4 tiles?"
- "Is the blue ship less then 4 blocks?"
- "Are all 3 ships the same size?"
- "Does the red ship have more blocks than the blue ship?"

As a result, the first key element of modeling question generation was to recognize the *compositionality* of these questions. In other words, there are conceptual building blocks (predicates like `size(x)` and `plus(x,y)`) that can be put together to create the meaning of other questions (`plus(size(Red), size(Purple))`). Combining meaningful parts to give meaning to larger expressions is a prominent approach in linguistics [10], and compositionality more generally has been an influential idea in cognitive science [4, 15, 14].

The second key element is the *computability* of questions. We propose that human questions are like programs that when executed on the state of a world output an answer. For example, a program that when executed looks up the number of blue tiles on a hypothesized or imagined Battleship game board and returns said number corresponds to the question "How long is the blue ship?". In this way, programs can be used to evaluate the potential for useful information from a question by executing the program over a set of possible or likely worlds and preferring questions that are informative for identifying the true world state. This approach to modeling questions is closely

related to formalizing question meaning as a partition over possible worlds [6], a notion used in previous studies in linguistics [18] and psychology [9]. Machine systems for question answering have also fruitfully modeled questions as programs [24, 12], and computational work in cognitive science has modeled various kinds of concepts as programs [17, 5, 13]. An important contribution of our work here is that it tackles *question asking* and provides a method for *generating* meaningful questions/programs from scratch.

## 4.2 A grammar for producing questions

To capture both compositionality and computability, we represent questions in a simple programming language, based on lambda calculus and LISP. Every unit of computation in that language is surrounded by parentheses, with the first element being a function and all following elements being arguments to that function (i.e., using prefix notation). For instance, the question "How long is the blue ship?" would be represented by the small program `(size Blue)`. More examples will be discussed below. With this step we abstracted the question representation from the exact choice of words while maintaining its meaning. As such the questions can be thought of as being represented in a "language of thought" [3].

Programs in this language can be combined as in the example `(> (size Red) (size Blue))`, asking whether the red ship is larger than the blue ship. To compute an answer, first the inner parentheses are evaluated, each returning a number corresponding to the number of red or blue tiles on the game board, respectively. Then these numbers are used as arguments to the > function, which returns either True or False.

A final property of interest is the *generativity* of questions, that is, the ability to construct novel expressions that are useful in a given context. To have a system that can generate expressions in this language we designed a grammar that is context-free with a few exceptions, inspired by [17]. The grammar consists of a set of rewrite rules, which are recursively applied to grow expressions. An expression that cannot be further grown (because no rewrite rules are applicable) is guaranteed to be an interpretable program in our language.

To create a question, our grammar begins with an expression that contains the start symbol A and then rewrites the symbols in the expression by applying appropriate grammatical rules until no symbol can be rewritten. For example, by applying the rules A → N, N → (size S), and S → Red, we arrive at the expression `(size Red)`. Table SI-1 (supplementary materials) shows the core rewrite rules of the grammar. This set of rules is sufficient to represent all 605 questions in the human data set.

To enrich the expressiveness and conciseness of our language we added lambda expressions, mapping, and set operators (Table SI-2, supplementary material). Their use can be seen in the question "Are all ships the same size?", which can be conveniently represented by `(= (map (λ x (size x)) (set Blue Red Purple)))`. During evaluation, map sequentially assigns each element from the set to x in the λ-part and ultimately returns a vector of the three ship sizes. The three ship sizes are then compared by the = function. Of course, the same question could also be represented as `(= (= (size Blue) (size Red)) (size Purple))`.

## 4.3 Probabilistic generative model

An artificial agent using our grammar is able to express a wide range of questions. To decide which question to ask, the agent needs a measure of question usefulness. This is because not all syntactically well-formed programs are informative or useful. For instance, the program `(> (size Blue) (size Blue))` representing the question "Is the blue ship larger than itself?" is syntactically coherent. However, it is not a useful question to ask (and is unlikely to be asked by a human) because the answer will always be `False` ("no"), no matter the true size of the blue ship.

We propose a probabilistic generative model that aims to predict which questions people will ask and which not. Parameters of the model can be fit to predict the frequency that humans ask particular questions in particular context in the data set by [19]. Formally, fitting the generative model is a problem of density estimation in the space of question-like programs, where the space is defined by the grammar. We define the probability of question $x$ (i.e., the probability that question $x$ is asked)

with a log-linear model. First, the *energy* of question $x$ is the weighted sum of question features

$$\mathcal{E}(x) = \theta_1 f_1(x) + \theta_2 f_2(x) + ... + \theta_K f_K(x), \tag{1}$$

where $\theta_k$ is the weight of feature $f_k$ of question $x$. We will describe all features below. Model variants will differ in the features they use. Second, the energy is related to the probability by

$$p(x; \boldsymbol{\theta}) = \frac{\exp(-\mathcal{E}(x))}{\sum_{x \in X} \exp(-\mathcal{E}(x))} = \frac{\exp(-\mathcal{E}(x))}{Z}, \tag{2}$$

where $\boldsymbol{\theta}$ is the vector of feature weights, highlighting the fact that the probability is dependent on a parameterization of these weights, $Z$ is the normalizing constant, and $X$ is the set of all possible questions that can be generated by the grammar in Tables SI-1 and SI-2 (up to a limit on question length).[3] The normalizing constant needs to be approximated since $X$ is too large to enumerate.

## 4.4 Optimization

The objective is to find feature weights that maximize the likelihood of asking the human-produced questions. Thus, we want to optimize

$$\arg \max_{\boldsymbol{\theta}} \sum_{i=1}^{N} \log p(d^{(i)}; \boldsymbol{\theta}), \tag{3}$$

where $D = \{d^{(1)}, ..., d^{(N)}\}$ are the questions (translated into programs) in the human data set. To optimize via gradient ascent, we need the gradient of the log-likelihood with respect to each $\theta_k$, which is given by

$$\frac{\partial \log p(D; \boldsymbol{\theta})}{\partial \theta_k} = N \, \mathbb{E}_{x \sim D}[f_k(x)] - N \, \mathbb{E}_{x \sim P_\theta}[f_k(x)]. \tag{4}$$

The term $\mathbb{E}_{x \sim D}[f_k(x)] = \frac{1}{N} \sum_{i=1}^{N} f_k(d^{(i)})$ is the expected (average) feature values given the empirical set of human questions. The term $\mathbb{E}_{x \sim P_\theta}[f_k(x)] = \sum_{x \in X} f_k(x) p(x; \boldsymbol{\theta})$ is the expected feature values given the model. Thus, when the gradient is zero, the model has perfectly matched the data in terms of the average values of the features.

Computing the exact expected feature values from the model is intractable, since there is a very large number of possible questions (as with the normalizing constant in Equation 2). We use importance sampling to approximate this expectation. To create a proposal distribution, denoted as $q(x)$, we use the question grammar as a probabilistic context free grammar with uniform distributions for choosing the re-write rules.

The details of optimization are as follows. First, a large set of 150,000 questions is sampled in order to approximate the gradient at each step via importance sampling.[4] Second, to run the procedure for a given model and training set, we ran 100,000 iterations of gradient ascent at a learning rate of 0.1. Last, for the purpose of evaluating the model (computing log-likelihood), the importance sampler is also used to approximate the normalizing constant in Eq. 2 via the estimator $Z \approx \mathbb{E}_{x \sim q}\left[\frac{p(x; \boldsymbol{\theta})}{q(x)}\right]$.

## 4.5 Question features

We now turn to describe the question features we considered (cf. Equation 1), namely two features for informativeness, one for length, and four for the answer type.

**Informativeness.** Perhaps the most important feature is a question's informativeness, which we model through a combination of Bayesian belief updating and Expected Information Gain (EIG). To compute informativeness, our agent needs to represent several components: A belief about the current world state, a way to update its belief once it receives an answer, and a sense of all possible

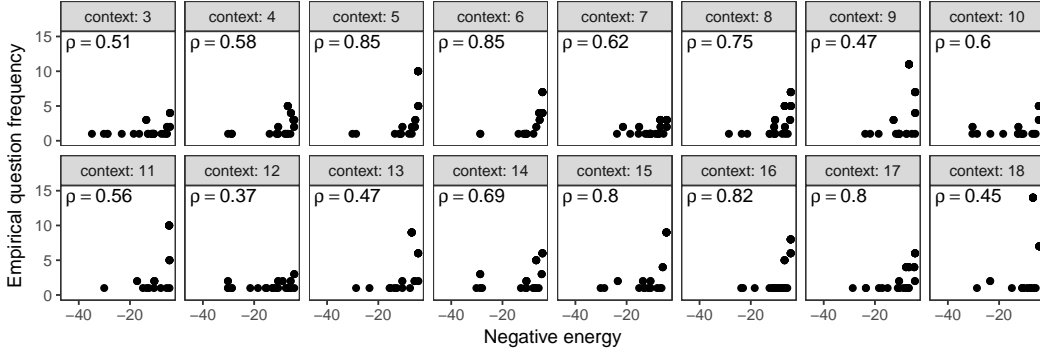

Figure 2: Out-of-sample model predictions regarding the frequency of asking a particular question. The y-axis shows the empirical question frequency, and x-axis shows the model's energy for the question (Eq. 1, based on the full model). The rank correlation $\rho$ is shown for each context.

answers to the question.[5] In the Battleship game, an agent must identify a single hypothesis $h$ (i.e., a hidden game board configuration) in the space of possible configurations $H$ (i.e., possible board games). The agent can ask a question $x$ and receive the answer $d$, updating its hypothesis space by applying Bayes' rule, $p(h|d;x) \propto p(d|h;x)p(h)$. The prior $p(h)$ is specified first by a uniform choice over the ship sizes, and second by a uniform choice over all possible configurations given those sizes. The likelihood $p(d|h;x) \propto 1$ if $d$ is a valid output of the question program $x$ when executed on $h$, and zero otherwise.

The Expected Information Gain (EIG) value of a question $x$ is the expected reduction in uncertainty about the true hypothesis $h$, averaged across all possible answers $A_x$ of the question

$$EIG(x) = \sum_{d \in A_x} p(d;x)\Big[I[p(h)] - I[p(h|d;x)]\Big], \tag{5}$$

where $I[\cdot]$ is the Shannon entropy. Complete details about the Bayesian ideal observer follow the approach we used in [19]. Figure 3 shows the EIG scores for the top two human questions for selected contexts.

In addition to feature $f_{\text{EIG}}(x) = \text{EIG}(x)$, we added a second feature $f_{\text{EIG=0}}(x)$, which is 1 if EIG is zero and 0 otherwise, to provide an offset to the linear EIG feature. Note that the EIG value of a question always depends on the game context. The remaining features described below are independent of the context.

**Complexity.** Purely maximizing EIG often favors long and complicated programs (e.g., polynomial questions such as `size(Red)+10*size(Blue)+100*size(Purple)+...`). Although a machine would not have a problem with answering such questions, it poses a problem for a human answerer. Generally speaking, people prefer concise questions and the rather short questions in the data set reflect this. The probabilistic context free grammar provides a measure of complexity that favors shorter programs, and we use the log probability under the grammar $f_{\text{comp}}(x) = -\log q(x)$ as the complexity feature.

**Answer type.** We added four features for the answer types Boolean, Number, Color, and Location. Each question program belongs to exactly one of these answer types (see Table SI-1). The type Orientation was subsumed in Boolean, with `Horizontal` as `True` and `Vertical` as `False`. This allows the model to capture differences in the base rates of question types (e.g., if people prefer true/false questions over other types).

**Relevance**. Finally, we added one auxiliary feature to deal with the fact that the grammar can produce syntactically coherent programs that have no reference to the game board at all (thus are not really questions about the game; e.g., `(+ 1 1)`). The "filter" feature $f_\emptyset(x)$ marks questions

that refer to the Battleship game board with a value of 1 (see the $^b$ marker in Table SI-1) and 0 otherwise.[6]

## 4.6 Alternative models

To evaluate which features are important for human-like question generation, we tested the full model that uses all features, as well as variants in which we respectively lesioned one key property. The *information-agnostic* model did not use $f_{\text{EIG}}(x)$ and $f_{\text{EIG}=0}(x)$ and thus ignored the informativeness of questions. The *complexity-agnostic* model ignored the complexity feature. The *type-agnostic* model ignored the answer type features.

# 5 Results and Discussion

The probabilistic model of question generation was evaluated in two main ways. First, it was tasked with predicting the distribution of questions people asked in novel scenarios, which we evaluate quantitatively. Second, it was tasked with generating genuinely novel questions that were not present in the data set, which we evaluate qualitatively. To make predictions, the different candidate models were fit to 15 contexts and asked to predict the remaining one (i.e., leave one out cross-validation).[7] This results in 64 different model fits (i.e., 4 models × 16 fits).

Table 1: Log likelihoods of model variants averaged across held out contexts.

| Model | LL |
|---|---|
| Full | -1400.06 |
| Information-agnostic | -1464.65 |
| Complexity-agnostic | -22993.38 |
| Type-agnostic | -1419.26 |

First, we verify that compositionality is an essential ingredient in an account of human question asking. For any given context, about 15% of the human questions did not appear in any of the other contexts. Any model that attempts to simply reuse/reweight past questions will be unable to account for this productivity (effectively achieving a log-likelihood of $-\infty$), at least not without a much larger training set of questions. The grammar over programs provides one account of the productivity of the human behavior.

Second, we compared different models on their ability to quantitatively predict the distribution of human questions. Table 1 summarizes the model predictions based on the log-likelihood of the questions asked in the held-out contexts. The full model – with learned features for informativeness, complexity, answer type, and relevance – provides the best account of the data. In each case, lesioning its key components resulted in lower quality predictions. The complexity-agnostic model performed far worse than the others, highlighting the important role of complexity (as opposed to pure informativeness) in understanding which questions people choose to ask. The full model also outperformed the information-agnostic and type-agnostic models, suggesting that people also optimize for information gain and prefer certain question types (e.g., true/false questions are very common). Because the log-likelihood values are approximate, we bootstrapped the estimate of the normalizing constant $Z$ and compared the full model and each alternative. The full model's log-likelihood advantage over the complexity-agnostic model held in 100% of the bootstrap samples, over the information-agnostic model in 81% of samples, and over type-agnostic model in 88%.

Third, we considered the overall match between the best-fit model and the human question frequencies. Figure 2 shows the correlations between the energy values according to the held-out predictions of the full model (Eq. 1) and the frequencies of human questions (e.g., how often participants asked "What is the size of the red ship?" in a particular context). The results show very strong agreement for some contexts along with more modest alignment for others, with an average Spearman's rank correlation coefficient of 0.64. In comparison, the information-agnostic model achieved 0.65, the complexity-agnostic model achieved -0.36, and the type-agnostic model achieved 0.55. One limitation is that the human data is sparse (many questions were only asked once), and thus correlations

are limited as a measure of fit. However, there is, surprisingly, no correlation at all between question generation frequency and EIG alone [19], again suggesting a key role of question complexity and the other features.

Last, the model was tasked with generating novel, "human-like" questions that were not part of the human data set. Figure 3 shows five novel questions that were sampled from the model, across four different game contexts. Questions were produced by taking five weighted samples from the set of programs produced in Section 4.4 for approximate inference, with weights determined by their energy (Eq. 2). To ensure novelty, samples were rejected if they were equivalent to any human question in the training data set or to an already sampled question. Equivalence between any two questions was determined by the mutual information of their answer distributions (i.e., their partitions over possible hypotheses), and or if the programs differed only through their arguments (e.g. `(size Blue)` is equivalent to `(size Red)`). The generated questions in Figure 3 demonstrate that the model is capable of asking novel (and clever) human-like questions that are useful in their respective contexts. Interesting new questions that were not observed in the human data include "Are all the ships horizontal?" (Context 7), "What is the top left of all the ship tiles?" (Context 9), "Are blue and purple ships touching and red and purple not touching (or vice versa)?" (Context 9), and "What is the column of the top left of the tiles that have the color of the bottom right corner of the board?" (Context 15). The four contexts were selected to illustrate the creative range of the model, and the complete set of contexts is shown in the supplementary materials.

## 6    Conclusions

People use question asking as a cognitive tool to gain information about the world. Although people ask rich and interesting questions, most active learning algorithms make only focused requests for supervised labels. Here were formalize computational aspects of the rich and productive way that people inquire about the world. Our central hypothesis is that active machine learning concepts can be generalized to operate over a complex, compositional space of programs that are evaluated over possible worlds. To that end, this project represents a step toward more capable active learning machines.

There are also a number of limitations of our current approach. First, our system operates on semantic representations rather than on natural language text directly, although it is possible that such a system can interface with recent tools in computational linguistics to bridge this gap [e.g., 24]. Second, some aspects of our grammar are specific to the Battleship domain. It is often said that some knowledge is needed to ask a good question, but critics of our approach will point out that the model begins with substantial domain knowledge and special purpose structures. On the other hand, many aspects of our grammar are domain general rather than domain specific, including very general functions and programming constructs such as logical connectives, set operations, arithmetic, and mapping. To extend this approach to new domains, it is unclear exactly how much new knowledge engineering will be needed, and how much can be preserved from the current architecture. Future work will bring additional clarity as we extend our approach to different domains.

From the perspective of computational cognitive science, our results show how people balance informativeness and complexity when producing semantically coherent questions. By formulating question asking as program generation, we provide the first predictive model to date of open-ended human question asking.

## Acknowledgments

We thank Chris Barker, Sam Bowman, Noah Goodman, and Doug Markant for feedback and advice. This research was supported by NSF grant BCS-1255538, the John Templeton Foundation Varieties of Understanding project, a John S. McDonnell Foundation Scholar Award to TMG, and the Moore-Sloan Data Science Environment at NYU.

## Context 7

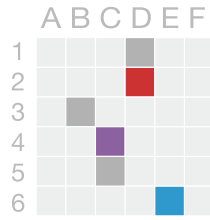

| EIG | Question/Program | Human |
|---|---|---|
| 2.44 | How many tiles are occupied by ships? <br> `(++ (map (lambda x (size x)) (set Blue Red Purple)))` | |
| 1.79 | How many ships are 4 tiles long? <br> `(++ (map (lambda x (== (size x) 4)) (set Blue Red Purple)))` | |

| Energy | Question/Program |
|---|---|
| 6.53 | What is the column of the bottom right water tile? <br> `(colL (bottomright (coloredTiles Water)))` |
| 7.88 | What is the row of the top left purple tile? <br> `(rowL (topleft (coloredTiles Purple)))` |
| 8.90 | Are all the ships horizontal? <br> `(all (map (lambda x (== H (orient x))) (set Blue Red Purple)))` |
| 10.51 | What is the column of the bottom right of the tiles with the same color as tile 3E? <br> `(colL (bottomright (coloredTiles (color 3E))))` |
| 12.89 | Are any of the ship sizes greater than 2? <br> `(any (map (lambda x (> (size x) 2)) (set Blue Red Purple)))` |

## Context 9

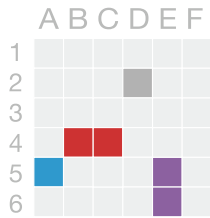

| EIG | Question/Program | Human |
|---|---|---|
| 1.59 | How many tiles in row 4 are occupied by ships? <br> `(++ (map (lambda y (and (== (rowL y) 4) (not (== (color y) Water)))) (set 1A ... 6F)))` | |
| 1.56 | How many tiles is the purple ship? <br> `(size Purple)` | |

| Energy | Question/Program |
|---|---|
| 7.48 | What is the column of the bottom right blue tile? <br> `(colL (bottomright (coloredTiles Blue)))` |
| 8.74 | How many tiles have the same color as tile 4A? <br> `(setSize (coloredTiles (color 4A)))` |
| 9.94 | What is the top left of all the ship tiles? <br> `(topleft (setDifference (set 1A ... 6F) (coloredTiles Water)))` |
| 10.98 | What is the color of the top left of the tiles that have the same color as 5C? <br> `(color (topleft (coloredTiles (color 5C))))` |
| 16.34 | Are blue and purple ships touching and red and purple not touching (or vice versa)? <br> `(== (touch Blue Purple) (not (touch Red Purple)))` |

## Context 15

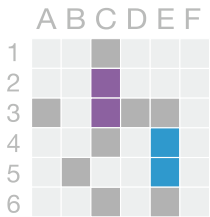

| EIG | Question/Program | Human |
|---|---|---|
| 4.09 | At what location is the top left part of the red ship? <br> `(topleft (coloredTiles Red))` | |
| 1.45 | How many tiles is the red ship? <br> `(size Red)` | |

| Energy | Question/Program |
|---|---|
| 7.23 | What is the row of the bottom right red tile? <br> `(rowL (bottomright (coloredTiles Red)))` |
| 7.36 | What is the row of the top left red tile? <br> `(rowL (topleft (coloredTiles Red)))` |
| 9.02 | How many tiles have the same color as tile 2F? <br> `(setSize (coloredTiles (color 2F)))` |
| 9.88 | What is the column of the top left of the tiles that have the color of the bottom right corner of the board? <br> `(colL (topleft (coloredTiles (color (bottomright (set 1A ... 6F))))))` |
| 11.45 | Is tile 5F a water tile? <br> `(isSubset (coloredTiles Water) (coloredTiles (color 5F)))` |

## Context 17

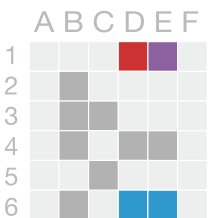

| EIG | Question/Program | Human |
|---|---|---|
| 2.24 | How many tiles in row 1 are occupied by ships? <br> `(++ (map (lambda y (and (== (rowL y) 1) (not (== (color y) Water)))) (set 1A ... 6F)))` | |
| 1.92 | At what location is the top left part of the red ship? <br> `(topleft (coloredTiles Red))` | |

| Energy | Question/Program |
|---|---|
| 6.53 | What is the column of the bottom right water tile? <br> `(colL (bottomright (coloredTiles Water)))` |
| 6.97 | What is the column of the top left water tile? <br> `(colL (topleft (coloredTiles Water)))` |
| 7.46 | How many tiles have the same color as the bottom right tile of the board? <br> `(setSize (coloredTiles (color (bottomright (set 1A ... 6F)))))` |
| 8.58 | What is the bottom right tile that has the same color as the tile 1A? <br> `(bottomright (coloredTiles (color (topleft (set 1A ... 6F)))))` |
| 9.08 | Are all the ships oriented horizontally? <br> `(all (map (lambda x (== H (orient x))) (set Blue Red Purple)))` |

Figure 3: Novel questions generated by the probabilistic model. Across four contexts, five model questions are displayed, next to the two most informative human questions for comparison. Model questions were sampled such that they are not equivalent to any in the training set. The natural language translations of the question programs are provided for interpretation. Questions with lower energy are more likely according to the model.

## Footnotes

[1] https://github.com/anselmrothe/question_dataset

[2]Although each of the 40 players asked a question for each context, a small number of questions were excluded from the data set for being ambiguous or extremely difficult to address computationally [see 19].

[3]We define $X$ to be the set of questions with 100 or fewer functions.

[4]We had to remove the rule L → (draw C) from the grammar and the corresponding 14 questions from the data set that asked for a demonstration of a colored tile. Although it is straightforward to represent those questions with this rule, the probabilistic nature of `draw` led to exponentially complex computations of the set of possible-world answers.

[5]We assume here that the agent's goal is to accurately identify the current world state. In a more general setting, the agent would require a cost function that defines the helpfulness of an answer as a reduced distance to the goal.

[6]The features $f_{\emptyset}(x)$ and $f_{\text{EIG}=0}(x)$ are not identical. Questions like `(size Blue)` do refer to the board but will have zero EIG if the size of the blue ship is already known.

[7]For computational reasons we had to drop contexts 1 and 2, which had especially large hypothesis spaces. However, we made sure that the grammar was designed based on the full set of contexts (i.e., it could express all questions in the human question data set).

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
