[Supplementary Material · rothe2017_nips_suppl_v1.pdf]

# Supplementary material: Question Asking as Program Generation

**Anselm Rothe**[1]
anselm@nyu.edu

**Brenden M. Lake**[1,2]
brenden@nyu.edu

**Todd M. Gureckis**[1]
todd.gureckis@nyu.edu

[1]Department of Psychology       [2]Center for Data Science
New York University

The supplementary material contains the following: the game boards that served as contexts in the human question data set (Figure SI-1), the full set of grammatical rules used in the simulations (Table SI-1 & SI-2), and five novel questions for each context produced by the computational model (Table SI-3 & SI-4).

Figure SI-1: Partly revealed game boards, serving as contexts in which participants generated questions from scratch in Rothe et al. (2016).

Table SI-1: Part 1 of the grammatical rules for defining the set of possible questions. Based on these rewrite rules we can represent all questions in the human question data set. See text for details. Rules marked with [b] have a reference to the Battleship game board (e.g., during evaluation the function *orient* looks up the orientation of a ship on the game board) while all other rules are domain-general (i.e., can be evaluated without access to a game board).

**Answer types**
A → B      *Boolean*
A → N      *Number*
A → C      *Color*
A → O      *Orientation*
A → L      *Location*

**Yes/no**
B → True
B → False
B → (not B)
B → (and B B)
B → (or B B)
B → (= B B)
B → (= N N)
B → (= O O)
B → (= C C)
B → (= setN)      *True if all elements in set of numbers are equal*
B → (any setB)      *True if any element in set of booleans is True*
B → (all setB)      *True if all elements in set of booleans are True*
B → (> N N)
B → (< N N)
B → (touch S S) [b]      *True if the two ships are touching (diagonal does not count)*
B → (isSubset setL setL)      *True if the first set of locations is subset of the second set of locations*

**Numbers**
N → 0
    ...
N → 10
N → (+ N N)
N → (+ B B)
N → (+ setN)
N → (+ setB)      *Number of True elements in set of booleans*
N → (− N N)
N → (size S) [b]      *Size of the ship*
N → (row L)      *Row number of location L*
N → (col L)      *Column number of location L*
N → (setSize setL)      *Number of elements in set of locations*

**Colors**
C → S      *Ship color*
C → Water
C → (color L) [b]      *Color at location L*
S → Blue
S → Red
S → Purple
S → x      *Lambda variable for ships*

**Orientation**
O → H      *Horizontal*
O → V      *Vertical*
O → (orient S) [b]      *Orientation of the ship S*

**Locations**
L → 1A      *Row 1, column A*
    ...
L → 6F
L → y      *Lambda variable for locations*
L → (topleft setL)      *The most left of the most top location in the set of locations*
L → (bottomright setL)      *The most right of the most bottom location in the set of locations*
L → (draw C)      *Sample a location of color C*

Table SI-2: Part 2 of the grammatical rules. See text for details.

| | |
|---|---|
| **Mapping** | |
| setB → (map fyB setL) | *Map a boolean expression onto location set* |
| setB → (map fxB setS) | *Map a boolean expression onto ship set* |
| setN → (map fxN setS) | *Map a numerical expression onto ship set* |
| setL → (map fxL setS) | *Map a location expression onto ship set* |
| | |
| **Lambda expressions** | |
| fyB → (λ y B) | *Boolean expression with location variable* |
| fxB → (λ x B) | *Boolean expression with ship variable* |
| fxN → (λ x N) | *Numeric expression with ship variable* |
| fxL → (λ x L) | *Location expression with ship variable* |
| | |
| **Sets** | |
| setS → (set Blue Red Purple) | *All ships* |
| setL → (set 1A ... 6F) | *All locations* |
| setL → (coloredTiles C) [b] | *All locations with color C* |
| setL → (setDifference setL setL) | *Remove second set from first set* |
| setL → (union setL setL) | *Combine both sets* |
| setL → (intersection setL setL) | *Elements that exist in both sets* |
| setL → (unique setL) | *Unique elements in set* |

Table SI-3: Part 1: Novel question programs generated by the probabilistic model. Model questions were sampled and filtered for novelty, meaning they never appeared in the training set. Please see main text for details of sampling process. The context ID refers to the contexts in Figure SI-1. The energy scores reflect the human-like "quality" assigned by the model.

| Context | Program | Energy |
|---|---|---|
| 3 | (colL (topleft (coloredTiles Water))) | 6.90 |
| 3 | (colL (bottomright (coloredTiles Red))) | 7.19 |
| 3 | (rowL (topleft (coloredTiles Red))) | 7.23 |
| 3 | (bottomright (coloredTiles (color 3A))) | 10.14 |
| 3 | (== Purple (color 2A)) | 10.20 |
| | | |
| 4 | (rowL (bottomright (coloredTiles Purple))) | 7.18 |
| 4 | (setSize (coloredTiles (color 1E))) | 8.65 |
| 4 | (== Purple (color 2D)) | 10.16 |
| 4 | (topleft (coloredTiles (color 2A))) | 10.70 |
| 4 | (color (topleft (map (lambda x 1C) (set Blue Red Purple)))) | 11.53 |
| | | |
| 5 | (setSize (coloredTiles (color (bottomright (set 1A ... 6F))))) | 7.55 |
| 5 | (setSize (coloredTiles (color 3A))) | 8.76 |
| 5 | (bottomright (coloredTiles (color 1B))) | 9.90 |
| 5 | (colL (bottomright (unique (coloredTiles Water)))) | 10.23 |
| 5 | (colL (bottomright (coloredTiles (color 3B)))) | 11.09 |
| | | |
| 6 | (colL (topleft (coloredTiles Water))) | 6.52 |
| 6 | (rowL (bottomright (coloredTiles Red))) | 7.33 |
| 6 | (setSize (coloredTiles (color 5A))) | 8.74 |
| 6 | (++ (map (lambda y (touch Blue Red)) (coloredTiles Water))) | 10.21 |
| 6 | (rowL (bottomright (coloredTiles (color 2A)))) | 11.30 |
| | | |
| 7 | (colL (bottomright (coloredTiles Water))) | 6.53 |
| 7 | (rowL (topleft (coloredTiles Purple))) | 7.88 |
| 7 | (all (map (lambda x (== H (orient x))) (set Blue Red Purple))) | 8.90 |
| 7 | (colL (bottomright (coloredTiles (color 3E)))) | 10.51 |
| 7 | (any (map (lambda x (> (size x) 2)) (set Blue Red Purple))) | 12.89 |
| | | |
| 8 | (rowL (bottomright (coloredTiles Red))) | 7.66 |
| 8 | (colL (bottomright (coloredTiles Red))) | 7.79 |
| 8 | (setSize (coloredTiles (color 2F))) | 9.09 |
| 8 | (++ (map (lambda y TRUE) (coloredTiles (color (topleft (set 1A ... 6F)))))) | 10.63 |
| 8 | (colL (topleft (coloredTiles (color (topleft (coloredTiles Blue)))))) | 10.79 |

Table SI-4: Part 2 of the novel question programs.

| Context | Program | Energy |
|---|---|---|
| 9 | (colL (bottomright (coloredTiles Blue))) | 7.48 |
| 9 | (setSize (coloredTiles (color 4A))) | 8.74 |
| 9 | (topleft (setDifference (set 1A ... 6F) (coloredTiles Water))) | 9.94 |
| 9 | (color (topleft (coloredTiles (color 5C)))) | 10.98 |
| 9 | (== (touch Blue Purple) (not (touch Red Purple))) | 16.34 |
| 10 | (rowL (bottomright (coloredTiles Blue))) | 7.10 |
| 10 | (colL (topleft (coloredTiles Purple))) | 7.15 |
| 10 | (rowL (topleft (coloredTiles Purple))) | 7.24 |
| 10 | (setSize (coloredTiles (color 4A))) | 8.69 |
| 10 | (bottomright (coloredTiles (color 4E))) | 10.27 |
| 11 | (colL (topleft (coloredTiles Red))) | 7.21 |
| 11 | (colL (bottomright (coloredTiles Red))) | 7.26 |
| 11 | (topleft (unique (coloredTiles Water))) | 9.18 |
| 11 | (topleft (coloredTiles (color 5F))) | 9.62 |
| 11 | (color (bottomright (coloredTiles (color 3E)))) | 10.64 |
| 12 | (colL (bottomright (coloredTiles Water))) | 6.65 |
| 12 | (colL (topleft (coloredTiles Water))) | 6.66 |
| 12 | (rowL (bottomright (coloredTiles Water))) | 7.35 |
| 12 | (setSize (coloredTiles (color 1A))) | 8.57 |
| 12 | (topleft (coloredTiles (color 1F))) | 9.91 |
| 13 | (colL (topleft (coloredTiles Water))) | 6.76 |
| 13 | (setSize (coloredTiles (color (bottomright (set 1A ... 6F))))) | 7.40 |
| 13 | (rowL (bottomright (coloredTiles Blue))) | 7.71 |
| 13 | (topleft (coloredTiles (color 4C))) | 9.99 |
| 13 | (colL (bottomright (coloredTiles (color 4E)))) | 11.09 |
| 14 | (colL (bottomright (coloredTiles Red))) | 7.59 |
| 14 | (setSize (coloredTiles (color 2A))) | 8.80 |
| 14 | (bottomright (coloredTiles (color (topleft (coloredTiles Water))))) | 9.72 |
| 14 | (topleft (coloredTiles (color 6F))) | 10.34 |
| 14 | (colL (bottomright (coloredTiles (color 5F)))) | 11.51 |
| 15 | (rowL (bottomright (coloredTiles Red))) | 7.23 |
| 15 | (rowL (topleft (coloredTiles Red))) | 7.36 |
| 15 | (setSize (coloredTiles (color 2F))) | 9.02 |
| 15 | (colL (topleft (coloredTiles (color (bottomright (set 1A ... 6F)))))) | 9.88 |
| 15 | (isSubset (coloredTiles Water) (coloredTiles (color 5F))) | 11.45 |
| 16 | (setSize (coloredTiles (color (topleft (set 1A ... 6F))))) | 7.83 |
| 16 | (colL (topleft (coloredTiles Red))) | 7.85 |
| 16 | (setSize (coloredTiles (color 2A))) | 8.85 |
| 16 | (topleft (coloredTiles (color 1D))) | 10.19 |
| 16 | (color (bottomright (setDifference (set 1A ... 6F) (coloredTiles Water)))) | 11.42 |
| 17 | (colL (bottomright (coloredTiles Water))) | 6.53 |
| 17 | (colL (topleft (coloredTiles Water))) | 6.97 |
| 17 | (setSize (coloredTiles (color (bottomright (set 1A ... 6F))))) | 7.46 |
| 17 | (bottomright (coloredTiles (color (topleft (set 1A ... 6F))))) | 8.58 |
| 17 | (all (map (lambda x (== H (orient x))) (set Blue Red Purple))) | 9.08 |
| 18 | (setSize (coloredTiles (color (topleft (set 1A ... 6F))))) | 7.41 |
| 18 | (bottomright (coloredTiles (color (topleft (set 1A ... 6F))))) | 8.79 |
| 18 | (setSize (coloredTiles (color 2D))) | 9.04 |
| 18 | (rowL (bottomright (coloredTiles (color 2A)))) | 10.76 |
| 18 | (colL (bottomright (coloredTiles (color 2D)))) | 11.18 |