[Reviews · NeurIPS 2017]

Reviewer 1



This paper attempts to reproduce user questions in a game of battleship where asking about individual squares is replaced by asking questions essentially over an artificial language with compositional logical (Montagovian) semantics. A log-linear model is used over a four simple features of the questions --- expected information gain (informativeness), answer type (Boolean or numerical), and whether the question involves features of the board at all. At a high level the motivation in terms of active learning is reasonable but the ideas in this paper are all rather obvious and the task is very simple, highly constrained, and logically unambiguous. I do not believe that this paper will be of interest to the NIPS community.

Reviewer 2



The authors examine human question asking where answers are K-ary. They define by hand a PCFG for a “battleship” domain, where there are hidden colored shapes in a partially observable grid (i.e., some tiles are revealed as containing part of a ship of a specific color or being empty). The task of the agent is to ask a question with a single word answer that provides as much information about the state of the board. The PCFG served as a prior over questions, which were defined as statements in lambda calculus. Question goodness was defined as a linear function of its informativeness (expected information gain), its complexity (in terms of its length or negative log probability under of it being generated by the PCFG with a uniform distribution over rewrite rules), and “answer type” (e.g., whether it provides a true/false or color as an answer). The human data came from a previously published paper and was hand coded into their grammar. The authors compared lesioned versions of the model via leave one out cross-validation and found that all factors provided a non-negligible contribution (although complexity was clearly the most important). Finally, they presented a series of novel questions. Question-asking is an important domain within linguistics and hypothesis testing. The authors provide an innovative perspective unifying modern linguistic and cognitive theory to examine how people ask questions, which has the potential to foster novel research in the active learning literature with models asking more sophisticated questions. The paper is extremely clear and well-written. The model is appropriately evaluated from both quantitative and qualitative perspectives. From the perspective of a computational cognitive scientist, this paper advances the field of self-directed learning. Previous work found that question informativeness was a poor predictor of human question asking. This followup paper resolves part of why this is the case (it is secondary to complexity and may in part be due to the size of possible questions available to the model). Thus, the paper contributes to the cognitive science literature, which may be sufficient to justify its publication in NIPS. However, there are serious limitations to their approach, which I can see limiting its interest to the more general NIPS audience. Although it may be possible to automate the human question text to statements in some semantic logic, it is unclear how the approach could scale or extend to domains of interest to the broader community. The battleship domain is small with only a few different features and their method is already computationally strained. That being said, the ideas are interesting enough (active learning with more rich questions) to be of interest, though it may be that the authors will need to provide a clear scenario of interest to the broader community where their technique is likely to be applicable and provide novel insights. Minor comments: 1) It may not be revealing, but I believe you can do a nested model comparison with the LLs of the lesioned models and get tests of statistical significance. 2) Moving forward, for sequential question asking, there are other models within the linguistics literature of question-asking in formal semantics and pragmatics. A lot of the work comes out of the Questions under Discussion literature. I listed a few references as pointers to this literature. Ginzburg, J. (1995). Resolving questions, part I. Linguistics and Philosophy, 18(5), 459-527. Rojas-Esponda, T. (2013). The roadsigns of communication. Proceedings of Semdial (DialDam), 17, 131-139.

Reviewer 3



The paper presents a cognitive model that is intended to predict when certain questions are asked in certain contexts (belief states). Under this model, the unnormalized log probability of a question is defined as the weighted sum of informativeness (expected information gain), complexity (roughly, question length), answer type, and "relevance" (whether the question refers to the belief state or just world-independent logical facts). This model is applied to the Battleship domain where the task is to figure out where within a grid a number of ships (clusters of grid elements) are located. The full model with all of the features mentioned above did better at predicting held-out human data (in terms of log-likelihood) than lesioned versions with fewer features. In terms of correlation with human judgments, the model that didn't take into account EIG did about as well as the full model. The paper seems generally technically sound. I think it succeeds more as an AI project than as a cognitive science project. The paper states that "We find that our model predicts what question people will ask" and "from a CogSci standpoint, our results show how people balance informativeness and complexity." However, I'm not convinced that we learn much about how people balance these factors. In the Battleship domain, it seems that people are either influenced by EIG a little (if we believe the log-likelihood results, which according to bootstrap estimates have a 19% chance of being wrong) or not at all (if we believe the correlation results, which are also pretty noisy due to sparse human data). I don't think these results allow us to conclude much about how people balance these factors in general. The paper is well-written and clear. I could probably reproduce the results. The paper is missing some prior work. One of the apparent innovations of the paper is to "model questions as programs that, when executed on the state of a possible world, output an answer." This is the same as the notion of a "goal" used in Hawkins 2015 ("Why do you ask? Good questions provoke informative answers."), and is based on the notion of a question-under-discussion (QUD) introduced by Roberts 1996 ("Information structure in discourse: Towards an integrated formal theory of pragmatics."). The questioner model by Hawkins also uses expected information gain and trades it off with the complexity of questions, but is applied to a much simpler domain. I think this sort of model (that has a compositional prior over questions and chooses based on EIG and other factors) makes a lot of sense as a component for cognitive and AI models, and I expect it to be used more in the future. I view the main contribution of this paper as showing that you can in fact generate interesting compositional questions for a non-trivial domain using this approach.